# LAM: LANGUAGE ARTICULATED OBJECT MODELERS

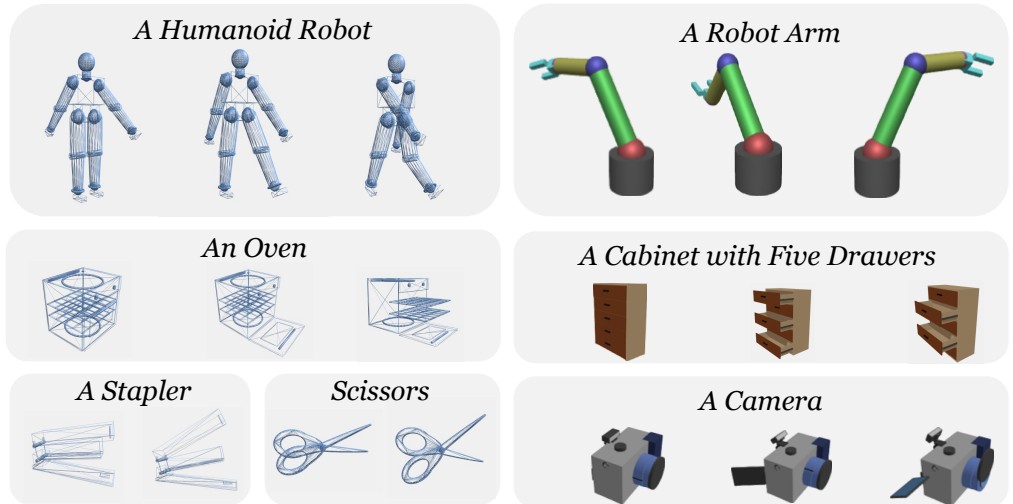

Figure 1: Our proposed pipeline can generate diverse articulated objects from text prompts, both without texture (left) and with texture (right). Users can easily control the articulations.

## ABSTRACT

We introduce LAM, a system that explores the collaboration of large-language models and vision-language models to generate articulated objects from text prompts. Our approach differs from previous methods that either rely on input visual structure (*e.g.*, an image) or assemble articulated models from pre-built assets. In contrast, we formulate articulated object generation as a unified code generation task, where geometry and articulations can be co-designed from scratch. Given an input text, LAM coordinates a team of specialized modules to generate code to represent the desired articulated object procedurally. The LAM first reasons about the hierarchical structure of parts (links) with Link Designer, then writes code, compiles it, and debugs it with Geometry & Articulation Coders and self-corrects with Geometry & Articulation Checkers. The code serves as a structured and interpretable bridge between individual links, ensuring correct relationships among them. Representing everything with code allows the system to determine appropriate joint types and calculate their exact placements more reliably. Experiments demonstrate the power of leveraging code as a generative medium within an agentic system, showcasing its effectiveness in automatically constructing complex articulated objects.

## 1 INTRODUCTION

Articulated objects are widespread in daily life, playing a crucial role in building realistic and interactive virtual environments for robotics, embodied AI, gaming, and VR/AR applications (Shen et al., 2021; Li et al., 2023; Ge et al., 2024; O'Neill et al., 2024; Liu et al., 2024a). Despite recent progress in simulation technology that significantly accelerates training through large-scale virtual environments (Xiang et al., 2020; Makoviychuk et al., 2021), the creation of articulated 3D assets remains a critical bottleneck. Unlike static 3D objects, which are abundantly available in large open-source datasets (Deitke et al., 2023b;a), articulated 3D models require expert manual annotation.

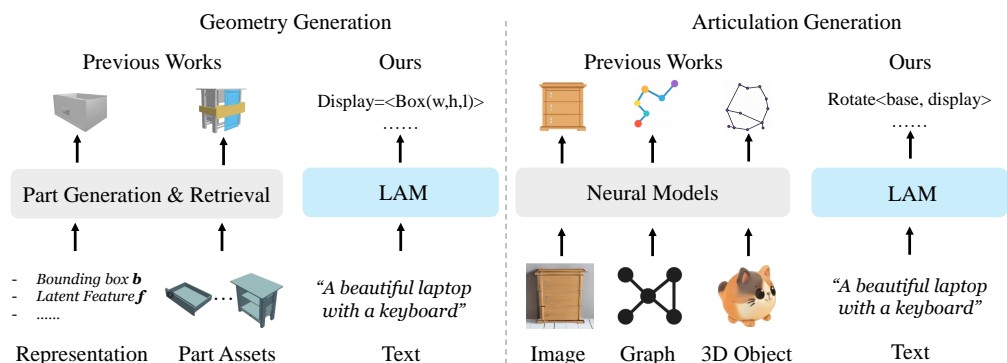

Figure 2: For geometry generation, previous works either rely on the 3D prior or retrieve pre-built 3D assets, the latter of which often leads to size mismatches as shown. For articulation generation, prior methods typically require an explicit arrangement representation as a learning medium, which imposes additional constraints on the range of possible articulation outcomes.

This is time-consuming, as complex objects are represented as hierarchical trees of parts and sub-parts (which are called *links* in this literature), along with corresponding joints, articulation types, and ranges of motion. This results in existing articulated object datasets having a relatively small scale (Mo et al., 2019; Liu et al., 2022). This limits the ability to leverage digital twins to train robots to interact with a broad variety of articulated objects. Automating the generation of articulation-ready models from textual descriptions represents a promising approach that we explore here to address this gap and enhance scalability in the creation of interactive virtual environments.

As shown in Figure 2, previous work on articulated object modeling has primarily relied on in-puts that contain structural information, such as images or videos (Mandi et al., 2024; Aygun & Mac Aodha, 2024; Yang et al., 2021; Song et al., 2024), graphs (Lei et al., 2023; Liu et al., 2024b), and meshes (Song et al., 2025; Qiu et al., 2025b), to reconstruct or generate objects with movable parts, often using predefined annotations and part graphs to guide the process. However, these methods are constrained by their reliance on structured data as input, which limits the diversity of producible articulated objects. Meanwhile, they cannot natively interpret abstract design descriptions and place parts without explicit structural guidance. In contrast, we introduce text-to-articulated-object genera-tion as a natural language interface that leverages large-scale language models to infuse semantic understanding into the generation process, thereby potentially reducing dependence on extensive 3D annotations and enabling more interactive and intuitive design iterations.

An articulated object consists of multiple parts (links) along with their corresponding 3D positions and connectivity relationships, which must be optimized simultaneously. Our key insight is to unify the complex, coupled problem of geometry and articulation generation into a single, expressive code representation. To manage this, our method — *LAM* — implements a collaborative framework where a team of specialized modules (composed by LLMs and 2D&3D VLMs) work together to generate a complete, articulated 3D object from a single text prompt. This process begins with **Link Designer** that reasons about the user's text to decompose the object into a hierarchical structure from shapes to parts to links and their relationships. Following this plan, **Geometry & Articulation Coders** translate the structure into executable code for both the precise geometry of each part and their kinematic joints. That code is checked by **Debuggers** for abnormalities. A cornerstone of our system is the automated, multi-modal feedback loop, which features **Geometry & Articulation Checkers** powered by 2D and 3D Vision-Language Models (VLMs). These modules render and analyze the current object design. Then, they provide targeted feedback, enabling the Coders to refine the code iteratively, ensuring the final model is both physically plausible and visually realistic before it is compiled.

The key contributions of our work include: (1) We introduce *LAM*, a collaborative system where a team of specialized agents (including **Designer**, **Coders**, **Debuggers**, and **Checkers**) generates articulated objects by operating on a unified code representation for both geometry and articulation. (2) We design an automated, multi-modal feedback system where 2D and 3D VLM-powered Checkers analyze rendered outputs to guide iterative code refinement, enabling self-correction without requiring pre-built assets or structural annotations. (3) Extensive experiments on the Part-Mobility dataset validate that our method achieves state-of-the-art performance in generation quality.

## 2 RELATED WORKS

**Articulated Objects Reconstruction**. Early methods train end-to-end models on synthetic data, simultaneously segmenting parts and predicting joint parameters through either interaction-based (Jiang et al., 2022; Hsu et al., 2023; Nie et al., 2022; Mu et al., 2021) or single-stage observations Heppert et al. (2023); Kawana et al. (2022); Wei et al. (2022). Per-object optimization techniques Liu et al. (2023b;a) avoid training but face scalability issues with multiple joints. Real2code Mandi et al. (2024) addresses this by leveraging LLMs to generate codes for each joint. Another line of work aims to predict articulation from pre-built meshes. Articulate AnyMesh Qiu et al. (2025a) and MagicArticulate Song et al. (2025) retrofit static meshes using VLMs and transformers, while IAAO Zhang & Lee (2025) enhances reconstruction via joint affordance prediction. Recent advances employ 3D Gaussian Splatting Kerbl et al. (2023). For example, ArticulatedGS Junfu et al. (2025) builds digital twins from multi-state point clouds, RigGS Yao et al. (2025) processes dynamic video input, and other works Yu et al. (2025); Wu et al. (2025); Kim et al. (2025) integrate visual-physical modeling with kinematic constraints.

**Articulated Objects Generation.** Diffusion-based methods have dominated recent advances. NAP Lei et al. (2023) utilizes graph-attention networks. CAGE Liu et al. (2024b) and ArtFormer Su et al. (2024) add user controllability for specifying constraints. Single-image generation also emerged as a key direction with SINGAPO Liu et al. (2025) learning plausible geometric variations, Phys-Part Luo et al. (2024) integrating physics constraints, and DreamArt Lu et al. (2025) employing three-stage pipelines with diffusion priors. Meanwhile, Infinite Mobility Lian et al. (2025) scales via procedural generation. Articulate-Anything Le et al. (2024) synthesizes Python code compiled to URDF, Real2Code Mandi et al. (2024) reconstructs up to 10 articulated parts via LLM-based code generation, and MeshArt Gao et al. (2025) employs hierarchical transformers for structured part-by-part generation.

In contrast, we introduce a collaborative system built upon a unified code representation that jointly models both object geometry and articulation. This integrated framework enables a closed-loop refinement process, allowing for the generation of physically plausible objects from text alone, without relying on the visual or structural priors required by previous methods.

## 3 LAM

### 3.1 PRELIMINARIES

**Representation of articulated objects.** We represent articulated objects using the Unified Robot Description Format (URDF), which encodes the geometry and kinematics of object parts, called *links*. Each link $L_i = \{\mathcal{M}_i, \mathbf{T}_i\}$ consists of a 3D mesh $\mathcal{M}_i$ and a pose $\mathbf{T}_i \in \mathrm{SE}(3)$, defined by its position $\mathbf{p}_i$ and roll-pitch-yaw (RPY) orientation $\boldsymbol{\theta}_i$. A joint $J_{pc}$ defines the kinematic connection between a parent link $L_p$ and a child link $L_c$. It is formally defined as $J_{pc} = (\mathbf{T}_{pc}, t_{pc}, \mathbf{a}_{pc}, \ell_{pc})$, where $\mathbf{T}_{pc} \in \mathrm{SE}(3)$ is the joint's pose relative to the parent, $t_{pc}$ is its type (e.g., `revolute`, `prismatic`), $\mathbf{a}_{pc} \in \mathbb{R}^3$ is the motion axis, and $\ell_{pc} = [\ell_{\min}, \ell_{\max}]$ are the motion limits. With the parent link $L_p$ at the origin, the child link's pose $\mathbf{T}_c$ is updated by the joint motion as:

$$\mathbf{T}_c^{'} = \mathbf{T}_p \cdot \mathbf{T}_{pc} \cdot \mathbf{X}(q_{pc}) \cdot \mathbf{T}_c, \tag{1}$$

where $\mathbf{X}(q_{pc}) \in \mathrm{SE}(3)$ is the joint transformation parameterized by the motion value $q_{pc}$ (e.g., rotation angle).

**Problem Formulation.** Given a textual description $x$, our goal is to generate an articulated object $\mathcal{A} = (\mathcal{L}, \mathcal{J})$. The object is composed of a *link set* $\mathcal{L} = \{L_i = (M_i, \mathbf{T}_i)\}_{i=1}^N$, containing $N$ meshes with corresponding poses, and a *joint set* $\mathcal{J} = \{J_{pc} = (\mathbf{T}_{pc}, t_{pc}, \mathbf{a}_{pc}, \ell_{pc})\}_{(p,c) \in \mathcal{E}}$, defining the kinematic connections. A compiler $\Psi$ then converts $\mathcal{A}$ into a collision-free and physically plausible URDF model $\mathcal{U} = \Psi(\mathcal{A})$.

### 3.2 ARTICULABLE GEOMETRY GENERATION

**Code-based Representation.** To make the structure of articulated objects tractable for LL, we introduce a hierarchical code-based representation progressing from *shape primitives* ($\mathcal{S}$) to *parts*

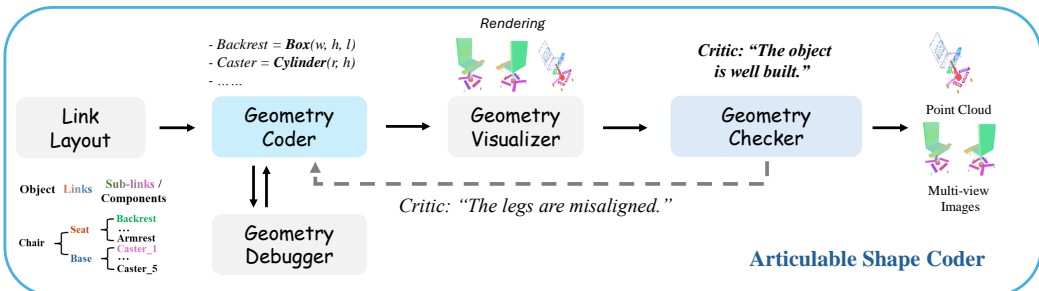

Figure 3: The overall framework of our proposed LAM. From a user's text prompt, LAM first designs a hierarchical structure of the object. It iteratively generates and refines code for both the geometry and articulation, resulting in an articulated object.

Figure 4: An overview of the Articulable Shape Coder. Given a hierarchical link layout output by Link Designer, our Geometry Coder generates code to define the shape and position of each link. Then, a VLM-powered Geometry Checker analyzes the rendered images and provides feedback, enabling an iterative refinement loop to correct geometric errors.

($\mathcal{P}$), and finally to *links* ($\mathcal{L}$). This structured representation circumvents the control limitations of end-to-end text-to-3D methods Shi et al. (2024); Long et al. (2024); Yan et al. (2024) and the oversimplification inherent in direct URDF generation. We define a set of parametric primitives, $\mathcal{S} = \{ s_k(\phi_k) \}_{k=1}^K$, built by calling functions like `<BoxGeometry>(l,w,h)` from the Three.js library. All primitives are normalized to a shared coordinate system for consistent alignment. Given a text instruction $x$, the **Geometry Coder** uses these primitive functions to generate shape primitives $\{ s_n(\phi_n) \}_{n=1}^N$, which can be hierarchically assembled into parts and then links. The final mesh geometry $\mathcal{M}_i$ and pose $\mathbf{T}_i$ for each link are thus defined within this program.

**Articulable Shape Generation with Iterative Refinement.** As illustrated in Figure 4, we frame the synthesis of link geometry and poses as a code-generation task orchestrated by LAM. Given an input text, the **Link Designer** (powered by an LLM) first reasons about the prompt to decompose the target object into a hierarchical structure of links and components. The **Geometry Coder** translates the generated link layout into executable code by selecting and parameterizing a library of predefined functions for both shape and pose. For shape generation, it employs primitive factory functions to instantiate and compose the mesh $\mathcal{M}_i$ for each link $L_i$. Concurrently, it determines the appropriate pose $\mathbf{T}_i$ (including position $\mathbf{p}_i$ and orientation $\boldsymbol{\theta}_i$) for each link. This methodology offers far greater control than generating raw URDF files or using text-to-3D models, thereby mitigating issues such as oversimplification or geometric uncontrollability. Usually, the initial code may contain geometric errors or physical implausibilities due to hallucinations. Therefore, we first employ **Geometry Debugger** to automatically fix grammar issues and then develop **Geometry Checker** to correct geometric errors, which is composed of 2D VLMs (*e.g.*, GPT-4o (Hurst et al., 2024)) or 3D VLMs (*e.g.*, PointLLM (Xu et al., 2024)). The **Geometry Visualizer** rendered multi-view images and a point cloud of the object (each link will be assigned a specific color for the Checker to refer to conveniently). Then, the **Geometry Checker** provides targeted feedback (*e.g.*, "The legs are misaligned") to enable an iterative refinement loop that corrects these errors. The final, validated link set, $\mathcal{L} = \{ L_i = (M_i, \mathbf{T}_i) \}_{i=1}^N$, forms the complete object geometry $\mathcal{A}$.

### 3.3 ARTICULATION GENERATION

Once the set of links $\mathcal{L} = \{ L_i = (\mathcal{M}_i, \mathbf{T}_i) \}_{i=1}^N$ is generated, the next crucial step is to define the kinematic joint set $\mathcal{J}$ that enables their articulation. This process is orchestrated by Articulation

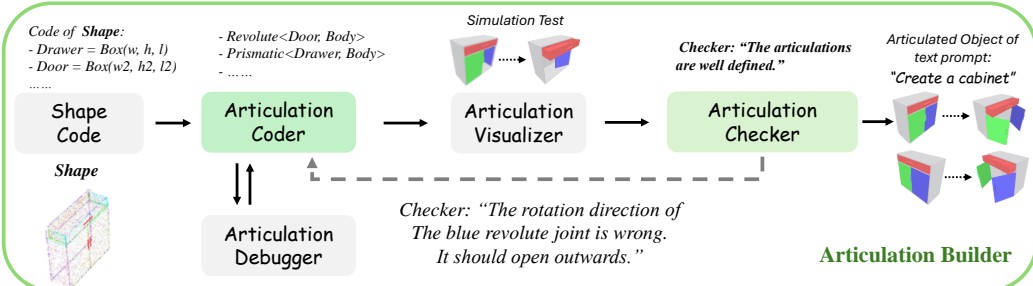

Figure 5: The Articulation Builder takes the generated shape code to define the object's articulation through a closed-loop process. An Articulation Coder generates code of joints, which the Articulation Visualizer then simulates to create a sequence of images to indicate the motion of joints. The Articulation Checker provides corrective feedback to iteratively refine the code until the motion is physically plausible and functionally correct.

Builder, as shown in Figure 5, which interprets the geometric and semantic properties of the links to produce a functionally correct articulation structure.

**Joint Assembly Solver.** Our approach first simplifies the complex problem of joint placement. Since the geometry generation stage (Section 3.2) produces links that are already well-aligned within a shared world matrix system, we bypass the need to predict complex relative joint poses. Instead, we focus on predicting the essential joint parameters: the joint type $t_{pc}$, the parent-child link pair $(L_p, L_c)$, and the absolute 3D position of the joint, $\mathbf{p}_{pc}$. The Articulation Builder achieves this by invoking the pre-defined meta-functions for formulating the articulable geometry to analyze the spatial relationships and functional affordances of the links based on their geometry ($\mathcal{M}_i$), pose ($\mathbf{T}_i$).

To correctly assemble the links according to the generated joint specifications, we introduce the **Joint Assemble Solver**, detailed in Algorithm 1. After designating a base link, the algorithm iterates through each joint. For `revolute` joints, it recalculates the child link's position to ensure it pivots correctly around the joint's position. The updated child position $\mathbf{p}_c^{\text{new}}$ is computed as $\mathbf{p}_c^{\text{new}} = \mathbf{p}_{pc} + \mathbf{R}_{pc}(\mathbf{p}_c - \mathbf{p}_{pc})$, where $\mathbf{R}_{pc}$ is the rotation matrix derived from the joint parameters. For `prismatic` and `fixed` joints, no position update is needed as their alignment is determined during geometry generation. Finally, any pose updates are recursively propagated down the kinematic chain.

---

**Algorithm 1. Joint Assemble Solver**

**Require:** Initial poses $\{\mathbf{T}_i\}$, Joint set $\mathcal{J} = \{J_{pc}\}$ with type $t_{pc}$, joint position $\mathbf{p}_{pc}$, and link poses $\mathbf{T}_p, \mathbf{T}_c$.
**Ensure:** Updated link poses $\{\mathbf{T}_i'\}$.
1: Designate a base link $L_{\text{base}}$.
2: **for** each joint $J_{pc}$ in $\mathcal{J}$:
3:     **if** $t_{pc}$ is `revolute`:
4:         Compute rotation matrix $\mathbf{R}_{pc}$
5:         $\mathbf{p}_c^{\text{new}} \leftarrow \mathbf{p}_{pc} + \mathbf{R}_{pc}(\mathbf{p}_c - \mathbf{p}_{pc})$.
6:         Update child pose $\mathbf{T}_c'$ based on $\mathbf{p}_c^{\text{new}}$.
7:     **else** (*prismatic* or *fixed*):
8:         Add $J_{pc}$ without pose changes.
9:     Recursively propagate pose updates for any subsequent joints connected to the updated $L_c$.

---

**Articulation Generation Using Shape Code with Checker.** As illustrated in Fig. 5, the generation and validation of the joint set $\mathcal{J}$ is performed through a closed-loop, multi-agent pipeline. Taking the generated shape code as input, the **Articulation Coder** generates executable code that defines the kinematic structure. It reasons about the object's components to establish parent-child hierarchies. It determines the appropriate joint type ($t_{pc}$), position ($\mathbf{p}_{pc}$), and motion axis ($\mathbf{a}_{pc}$) for each connection. Concurrently, a **Articulation Debugger** collaborates to resolve any syntax or code-level errors, ensuring the generated script is valid. The validated code is then passed to the **Articulation Visualizer**. To enable the **Articulation Checker** to provide targeted feedback, the **Articulation Visualizer** assigns a unique color to the child link of each joint. The corresponding mapping between colors and link semantics is then passed to the 2D VLM-powered **Articulation Checker**. It assesses the functional plausibility of the object's movement. For instance, it can detect if a cabinet door opens in the wrong direction or if a drawer's movement is unnatural (as shown in Figure 5). Based on its assessment, it provides feedback (*e.g.*, "The rotation direction of the blue revolute joint is wrong. It should open outwards."). This feedback guides the **Articulation Coder** to refine the code iteratively. This loop continues until the critic

Table 1: Quantitative comparisons on the success rate of text-based joint prediction. (a) To fairly compare our method with the Real2Code, we use the 5 classes from their paper: `Laptop`, `Box`, `Refrigerator`, `Storage-Furniture`, and `Table` categories for comparison. (b) URDF-former, Articulate Anything, and LAM (ours) support any number of classes; results here are for all 40 classes of the Part-Mobility dataset.

| (a) Results on Five classes | | (b) Results on *General Classes* | |
|---|---|---|---|
| Method | Success Rate | Method | Success Rate |
| Real2Code | 13.5% | URDFormer | 14.6% |
| Articulate Anything | 40.3% | Articulate Anything | 48.9% |
| LAM (ours) | **77.1**% | LAM (ours) | **63.7**% |

confirms that the articulations are well-defined and physically correct, resulting in the final, validated joint set $\mathcal{J}$.

# 4 EXPERIMENT

**Datasets.** To ensure a fair comparison with prior works (Liu et al., 2025; Su et al., 2024), we conduct evaluations on the same subsets of the Part-Mobility dataset as the prior papers (5 classes for Mandi et al. (2024); 6 classes for Su et al. (2024)). Furthermore, to provide a more comprehensive analysis of our method's capabilities in generating diverse articulated objects, we also extend our experiments to include all 46 object categories available in the Part-Mobility dataset, referred to as *General Classes*. For each category, we use the official rendered images to generate one caption per category. Meanwhile, we also collect a more challenging set of 27 descriptions of complex articulated objects, noted as *Open-World Classes*. The descriptions can be found in the Appendix A.5.

**Benchmark and Metrics.** We first adopt a masked URDF reconstruction task to validate joint placement ability and evaluate the success rate as defined in work (Le et al., 2024). We also measure geometric quality and diversity using **Minimum Matching Distance (MMD)**, **Coverage (COV)**, and **1-Nearest Neighbor Accuracy (1-NNA)** (Su et al., 2024; Liu et al., 2024b). Text-to-image alignment is quantified via CLIP (Radford et al., 2021) and BLIP (Li et al., 2022) scores. For automated evaluation, GPT-4o (Lin et al., 2024) performs articulation examinations and pairwise preference comparisons. Finally, we use the accuracy of the generated articulated objects (both the links and the articulations should be correct) of the collected 83 captions to ablate the variant designs of LAM.

**Implementation Details.** Our framework centrally employs LLMs and VLMs for generating the code that defines object geometry and articulation. The Linker Designer is implemented by GPT-4o. For the Articulable Geometry Generation, we use Gemini-2.5-pro and functions defined from the Three.js library by default. We use o3 equipped with the proposed Joint Assembly Solver as Articulation Coder. Geometry & Articulation Checkers are based on the Gemini-2.5-flash and PointLLM (Xu et al., 2024). The Debuggers are also Gemini-2.5-flash with deterministic Python & JavaScript scripts to verify the issues. More details of each module are listed in the Appendix A.7.

## 4.1 MAIN RESULTS

**Success Rate Comparison of Joint Prediction**. In Table 1, on the dataset classes from Real2Code, our LAM model achieves a success rate of 77.1%, which significantly surpasses both Articulate Anything (40.3%) and Real2Code (13.5%). This robust performance is consistent even on the more diverse General Classes, where LAM attains a 63.7% success rate, again outperforming the strongest baseline, Articulate Anything (48.9%). These experiments validate the superior capability of our proposed method in accurately predicting and placing joints based on textual descriptions.

**Visual Alignment and Generation Quality Comparisons.** Table 2 presents a comprehensive evaluation of our LAM model against several baselines, assessing both the visual-semantic alignment with text prompts and the quality of in-distribution generation. In the visual alignment and articulation preference comparisons, our method demonstrates clear superiority. LAM achieves the highest CLIP and BLIP scores (31.94 and 63.76, respectively), indicating a stronger semantic correspondence between the generated 3D objects and the input text compared to CAGE, SINGAPO, and Articulate

Table 2: Quantitative comparisons on (`Storage Furniture`, `Table`, `Refrigerator`, `Dishwasher`, `Oven`, and `Washer`), which are the shared classes among CAGE and Singapo. **(a)** Visual alignment (CLIP, BLIP scores; higher is better) and articulation modeling (GPT-4o pass rate). **(b)** In-distribution generation quality using MMD (lower is better), COV (higher is better), and 1-NNA (lower is better) metrics. ArtFormer-PR means ArtFormer framework with part retrieval.

<table>
<tr><td colspan="4">(a) Visual alignment and GPT-4o pass rate.</td><td colspan="4">(b) Generation quality Comparisons.</td></tr>
<tr><td>Method</td><td>CLIP ↑</td><td>BLIP ↑</td><td>GPT-4o ↑</td><td>Method</td><td>MMD ↓</td><td>COV ↑</td><td>1-NNA ↓</td></tr>
<tr><td>CAGE</td><td>27.65</td><td>53.92</td><td>58.8%</td><td>CAGE</td><td>0.0193</td><td>0.6064</td><td>0.5319</td></tr>
<tr><td>SINGAPO</td><td>30.43</td><td>56.21</td><td>61.4%</td><td>ArtFormer</td><td>0.0292</td><td>0.5213</td><td>0.5266</td></tr>
<tr><td>Articulate Anything</td><td>28.23</td><td>56.99</td><td>70.2%</td><td>ArtFormer-PR</td><td>0.0214</td><td>0.6400</td><td>0.3950</td></tr>
<tr><td>LAM (Ours)</td><td>31.94</td><td>63.76</td><td>78.6%</td><td>LAM (Ours)</td><td>0.0149</td><td>0.6871</td><td>0.3599</td></tr>
</table>

Anything. Furthermore, our model achieves a GPT-4o pass rate of 78.6%, indicating that its generated articulations are overwhelmingly considered functionally correct and plausible, substantially outperforming all baselines. For in-distribution generation quality, our approach continues to excel, achieving the best performance across all standard metrics. It records the lowest MMD (0.0149) and 1-NNA (0.3599), which confirms that the distribution of our generated shapes is closer to the ground-truth data and more realistic. Concurrently, LAM scores the highest in COV (0.6871), reflecting its capability to produce a more diverse set of objects that better covers the data manifold. These combined results underscore the effectiveness of our code-based framework in producing not only visually and semantically accurate but also high-quality and diverse articulated objects.

**Comparisons on General Classes**. As shown in Figure 6, our LAM model demonstrates substantially better performance than Articulate Anything on both General and the more challenging Open-World object classes. For General Classes, LAM achieves significantly higher visual-semantic alignment with CLIP and BLIP scores of 31.21 and 58.94, respectively, compared to the baseline's 25.34 and 48.32. More importantly, it garners an overwhelming preference from both GPT-4o (81.1%) and human users (84.6%). These strong preference rates from both automated and human evaluators underscore that the objects generated by LAM are not only semantically aligned but also perceived as more functionally plausible and visually coherent. This performance gap widens in the Open-World evaluation, where LAM's user preference score reaches 91.7%, showcasing its superior generalization and ability to generate plausible articulated objects from diverse, unseen text prompts.

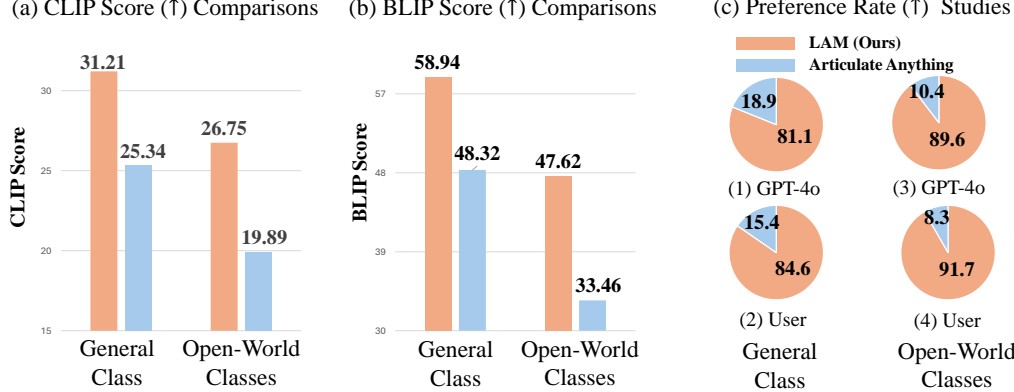

Figure 6: System-level comparisons for General and Open-World classes. For open-world classes, we collect a list of text descriptions about diverse articulated objects in the world, such as `Ferris wheel`, `shutter`, etc. (a) LAM achieves the best CLIP score on both General Classes and the new Open-World Classes. (b) LAM also achieves the best BLIP scores. (c) Both GPT-4o and human participants in our user study prefer the objects (given simulated videos to show motion) generated by LAM over those generated by Articulate Anything.

Table 3: Ablation Studies on the effect of Checkers and their designs. Multi-view refers to using four images rather than one image for the Geometry Checker to provide feedback. Image Sequence means using multiple intermediate motion statuses to pass to the Articulation Checker to get feedback.

(a) Effects of Checkers

| Geometry Checker | Articulation Checker | Max Iter | Acc. ↑ |
|:---:|:---:|:---:|:---:|
| ✗ | ✗ | - | 50.6% |
| ✓ | ✗ | - | 61.4% |
| ✗ | ✓ | 1 | 56.6% |
| ✓ | ✓ | 1 | 66.3% |
| ✓ | ✓ | 3 | **75.9%** |

(b) Effects of the design of Checkers

| Geometry Checker Type | Multi-View | Images Sequence | Acc. ↑ |
|:---:|:---:|:---:|:---:|
| 2D | ✗ | ✗ | 60.2% |
| 2D | ✓ | ✗ | 65.1% |
| 2D | ✓ | ✓ | 71.1% |
| 3D | ✓ | ✓ | 62.7% |
| 2D & 3D | ✓ | ✓ | **75.9%** |

## 4.2 ABLATION STUDIES

We utilize the combination of captions from General Classes from the Part-Mobility dataset and self-collected descriptions of Open-World Classes to evaluate the performance of different settings, resulting in a total of 83 classes. For each category, I generate one object per class for validation. We use accuracy (Acc.) to judge each setting, which means the generated objects should at least include the correct shape layout and joints with accurate placements.

**Effects of Checkers.** As shown in Table 3a, our proposed Geometry & Articulation Checkers are vital. The baseline accuracy without any Checker is 50.6%. Introducing the Geometry Checker or Articulation Checker alone improves accuracy to 61.4% and 56.6%, respectively. Employing them together raises the accuracy to 66.3%, indicating their complementary roles. Increasing the refinement iterations to three achieves the highest accuracy of 75.9%, which highlights the effectiveness of the iterative feedback loop in generating plausible objects.

**Effects of the design of Checkers.** Table 3b shows the impact of Checker design choices. A basic 2D Checker using a single image yields 60.2% accuracy. This increases to 65.1% when using multi-view images and further to 71.1% with the addition of image sequences to evaluate motion. While a 3D-only Checker is less effective (62.7%), a hybrid approach combining both 2D and 3D Checkers achieves the best performance at 75.9%. This suggests that 2D and 3D Checkers provide complementary feedback, making their combination the most effective configuration.

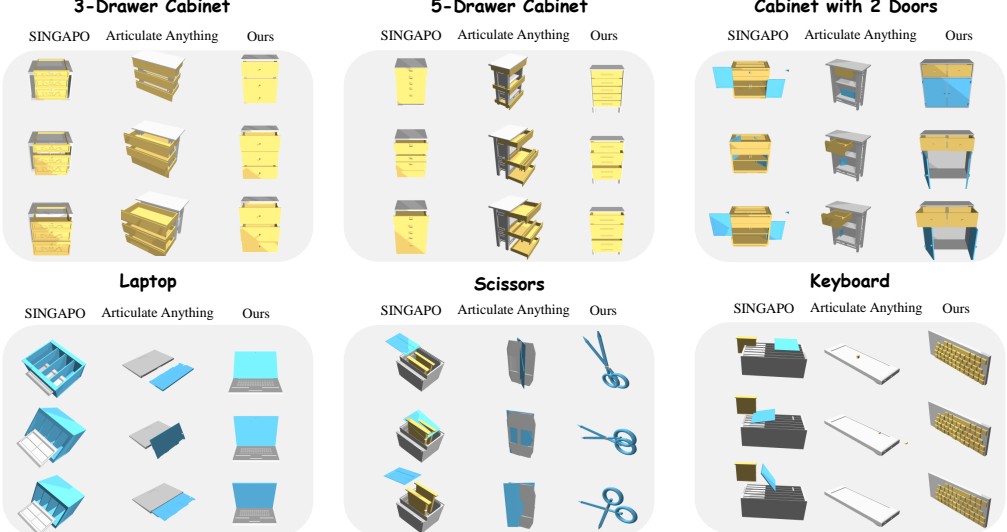

Figure 7: Six examples, where only the Cabinet classes is ID for SINGAPO, illustrating generation quality across different difficulty levels. Not unexpectedly, SINGAPO fails to produce sensible objects on the OOD classes. Articulate Anything also struggles on keyboard, laptop and scissors.

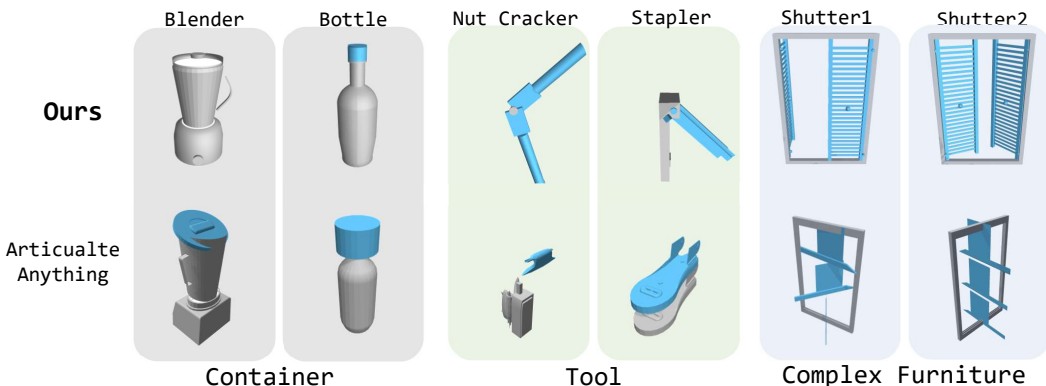

Figure 8: **Open-Vocabulary Scenarios.** Our model consistently outperforms Articulate Anything.

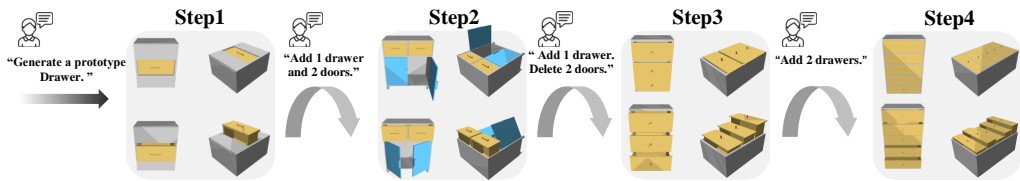

Figure 9: **Instruction-following ability.** In four steps, including adding and removing sub-objects, a one-drawer cabinet is guided to be a five-drawer cabinet.

### 4.3 QUALITATIVE RESULTS

**Overall qualitative comparisons**. Figure 7 illustrates our method's performance across six diverse zero-shot targets: simple (3- and 5-drawer cabinets), moderate (laptop, high-end cabinet), and OOD (keyboard, scissors). Our pipeline successfully encodes each link as a precisely posed URDF mesh and accurately predicts all joints. The output is always collision-free and correctly articulated, whereas Singapo and Articulate Anything frequently misplace parts or omit hinges and keys. The combination of stability on simpler tasks, excellent visual quality on more challenging ones, and strong generalization to OOD examples clearly demonstrates the superiority of our approach.

**Open-Vocabulary Scenarios**. Figure 8 compares our model with Articulate Anything across three domains—containers (spatial reasoning), tools (precision), and complex furniture (structural complexity). Our system shows stronger command understanding and physical common sense: it tracks part-to-part spatial relations more accurately, identifies movable or interactive components more explicitly, and handles highly intricate, mesh-like structures and dense layouts.

**Instruction-following Ability**. Integrating high-context LLMs into our pipeline makes the system portable and reusable, chiefly by enabling instruction following. Prior outputs can feed later stages, so the model refines its own work—cutting users' descriptive burden, supporting incremental edits of complex objects, and allowing repeated iterations. Figure 9 shows that in four steps (including adding and removing), a one-drawer cabinet can be instructed to become a five-drawer cabinet.

## 5 CONCLUSION

We introduced LAM, a pioneering system that generates articulated 3D objects from text by unifying geometry and articulation within a single code representation. Our framework uniquely employs a collaborative team of specialized AI modules—including Designers, Coders, and Checkers—to iteratively write, debug, and refine this code through a closed-loop, multi-modal feedback process. Extensive experiments demonstrate that LAM significantly surpasses previous methods in generation quality, text alignment, and diversity, particularly showcasing robust generalization on challenging open-world classes. By streamlining the creation of articulation-ready assets, LAM offers a promising solution for applications in robotics, VR/AR, and simulation.

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

# A APPENDIX

## CONTENTS OF THIS APPENDIX

Link Designer – Geometry Coder – Geometry Debugger – Geometry Visualizer – Geometry Checker – Articulation Coder.

More details of the used metrics.

Analysis of Failure Cases – More Comparisons with Previous works.

## A.1 ETHICS OF STATEMENT

The LAM system offers significant societal benefits by simplifying the creation of articulated 3D assets, which are essential in fields like robotics, embodied AI, gaming, and virtual or augmented reality. Making asset creation more accessible could democratize content production and enhance the diversity of interactive objects available for AI training. Nonetheless, generative AI technologies such as LAM come with certain risks. These include potential misuse, such as generating deceptive or misleading content. Furthermore, biases present in training datasets might inadvertently be perpetuated, and automation facilitated by such technologies may result in job displacement in creative industries.

## A.2 REPRODICIBILITY STATEMENT

To ensure the reproducibility of our research, we have provided the complete source code for our proposed LAM pipeline in the supplementary material. Our experiments are primarily conducted on the publicly available Part-Mobility dataset. For our open-world evaluations, the complete list of text descriptions used is also available in the appendix, allowing for a comprehensive replication of our results. Our framework is built upon large-scale language and vision-language models that are publicly accessible via APIs, including GPT-4o, Gemini-2.5-pro, and PointLLM. We believe that the combination of our provided code, the public dataset, and the detailed model specifications will enable the research community to verify and build upon our work.

## A.3 USE OF LLMS

In the development and preparation of this research paper, Large Language Models (LLMs) served as a valuable assistive tool. During the implementation phase of our project, we utilized LLMs to aid in debugging our codebase and to accelerate the development process. Furthermore, for the manuscript itself, an LLM was employed to perform grammar and syntax checks, thereby enhancing the overall clarity and readability of the text. It is important to note that this application of LLMs is limited to the development and writing process, and is distinct from the role of LLMs as a core component of our proposed methodology.

## A.4 LIMITATIONS

While our method represents a notable advancement in generating articulated objects from text, it faces certain limitations. The reliance on a predefined set of geometric primitives constrains the generation of highly detailed and intricate shapes, limiting its suitability for applications that demand fine-grained precision. Beyond geometry, accurately capturing complex kinematics remains a significant challenge. Even when individual parts are well-formed, the model can produce subtle inaccuracies in joint definitions, particularly for objects with multiple degrees of freedom or unconventional articulation mechanisms. These errors often manifest as plausible yet functionally incorrect joint axis orientations, motion ranges, and movement directions, indicating a need for more nuanced kinematic reasoning.

## A.5 COLLECTED 27 DESCRIPTIONS OF OPEN-WORLD CLASSES

1: A Ferris Wheel;
2: A bicycle wheel;
3: A Robot Arm consists of a Base (fixed or mobile), a series of rigid Links (segments), and Joints connecting them, terminating in an End Effector (gripper, tool);
4: A Tripod has three adjustable Legs connected to a central Head/Mounting Plate;
5: A shutter;
6: A bi-fold closet door system;
7: A four-wheeled golf cart with bag storage compartment;
8: A shopping cart;
9: A blender;
10: Portable folding chair;
11: A bicycle;
12: A common nutcracker design uses two rigid Lever Arms joined at one end by a Hinge/Pivot;
13: A Car Door consists of the main Door Panel (outer skin, inner panel, window frame);
14: A spring-type Clothespin consists of two identical Lever Arms (wood or plastic);
15: An Action Figure represents a character with multiple points of articulation (joints) connecting body parts like Head, Torso, Upper Arms, Forearms, Hands, Upper Legs, Lower Legs, Feet;
16: A Bicycle Chain is composed of many interconnected Links. Each link consists of Inner Plates, Outer Plates, Pins, and Rollers;
17: A Gate Leg Table has a fixed Top Center Section and one or two hinged Leaves (Side Sections);
18: A Metal Link Watch Band consists of numerous small, interconnected metal Links that articulate to conform to the wrist.;
19: A Makeup Compact is typically a small, flattened case (often round or square) with a hinged Lid.;
20: Retractable patio awning;
21: A piano;
22: A bookshelf;
23: A Caliper;
24: A mobile crane with telescopic boom extension;
25: A crimping tool;
26: An excavator arm;
27: A professional hydraulic jack with safety valve and wide base;

## A.6 TEXTURE GENERATION

In addition to geometry and articulation, the LAM framework includes an optional module for programmatic texture generation to enhance the visual realism of the final objects. As shown in the overall framework (Figure 3), this process is initiated after the articulable geometry is finalized.

To achieve this, we employ a *Texture Generator* module, which is powered by a large language model (LLM) such as Gemini-2.5-pro. This module is tasked with generating `three.js` code to define the material properties for each link. The generated code specifies the material type (e.g., `MeshStandardMaterial`) and its associated parameters, such as color, roughness, and metalness, tailored to the object's components. This code is then executed to render the object with the specified textures before being exported.

While this module allows for the creation of high-fidelity, textured assets, it remains an optional step within our pipeline. To ensure a fair and direct comparison with prior works, all quantitative experiments and results reported in the main body of this paper were conducted on objects generated without textures.

### A.7 MORE DETAILS OF EACH MODULE

More technical details for the main modules are listed here. For the prompt of each module, please refer to the code in the supplementary material.

#### A.7.1 LINK DESIGNER

This module is the foundational module in the LAM framework, tasked with interpreting a user's text prompt and decomposing the target object into a hierarchical structure of its constituent parts, known as **links**. This process results in a structured link layout, typically formatted as a JSON tree, which serves as a comprehensive blueprint for the downstream **Coders** and **Builders**.

To accurately represent kinematic relationships, the module organizes the object's components into a clear hierarchy that naturally encodes the parent-child relationships essential for defining the kinematic chain. Each component within this tree is annotated with descriptions for its geometry (shaping) and its spatial relationship to other components (positioning). To ensure the process remains tractable for highly complex objects, the Link Designer intelligently aggregates repetitive elements, such as the casters on an office chair or the keys on a keyboard, into single logical groups. This structured link layout is then passed to the subsequent modules in the pipeline. The **Articulable Geometry Coder** uses the geometric and positional descriptions to generate executable code defining the 3D mesh ($M_i$) and pose ($T_i$) for each individual link. Following that, the **Articulation Builder** leverages the same hierarchy and semantic information to infer and generate code for the **joints** that connect these links. This modular approach, which hinges on the structured output from the Link Designer, ensures that the coupled problem of geometry and articulation generation is grounded in a unified and coherent plan derived directly from the initial text prompt.

#### A.7.2 GEOMETRY CODER

The Geometry Coder transforms the hierarchical link layout, as specified by the Link Designer, into executable Three.js code for 3D mesh generation. This module is designed to convert abstract structural descriptions into geometrically valid 3D models, ensuring that the output is organized into articulation-ready groups.

The coder leverages a comprehensive Three.js geometry library including 15 primitive types (Box-Geometry, CylinderGeometry, ExtrudeGeometry, LatheGeometry) and advanced operations (CSG boolean operations, Matrix4 transformations). Complex shapes are constructed through hierarchical composition—a laptop hinge might combine CylinderGeometry for the pivot, BoxGeometry for mounting brackets, and TorusGeometry for washers. The coder averages 8.3 primitives per link, balancing geometric fidelity with computational efficiency. Then, the coder processes the hierarchical structure from the Link Designer, implementing a strict mapping policy: parent link nodes become `THREE.Group` containers, while child components become meshes within their parent groups. This preserves kinematic relationships—components that articulate together remain in the same group, enabling proper transformation propagation. This grouping strategy reduces the number of exported components from potentially 100+ individual meshes to 10–20 logically organized groups.

#### A.7.3 GEOMETRY DEBUGGER

The Geometry Debugger is a specialized module designed to address a critical inefficiency in the iterative generation process: syntax and grammar errors in the Three.js code produced by the Geometry Coder. Instead of resorting to a computationally expensive full regeneration of the code, this module employs lightweight LLMs (e.g., `gemini-2.5-flash`) to perform targeted repairs. This approach significantly reduces both cost and latency while preserving the geometric integrity of the object's links. To handle variability in LLM output formats, the Geometry Debugger utilizes a multi-tier extraction hierarchy to robustly parse the corrected code from the model's response. Following extraction, a dual validation pipeline is executed. This combines automated syntax checking using a Node.js subprocess with heuristic validation that checks for delimiter balance, import consistency, and correct function patterns. Rather than attempting a single-shot fix, the debugger engages in an incremental refinement loop. If a fix attempt fails, the resulting error message is fed back into the context for the next attempt, allowing the model to learn from its previous failures within the same session. Throughout this process, explicit instructions are provided to avoid modifying shape

parameters, ensuring that the geometric definitions remain faithful to the Geometry Coder's original output.

### A.7.4 GEOMETRY VISUALIZER

The Geometry Visualizer module transforms the executable code generated by the Geometry Coder into multi-modal visual representations—multi-view images and a point cloud—for analysis by the Geometry Checker. The process begins by orchestrating the transformation from Three.js code to OBJ meshes within a headless Node.js execution environment, which features dynamic ES module path resolution and regex-based error pattern extraction to provide targeted feedback on code-level issues. The core contribution for visual analysis is link-based semantic coloring; instead of coloring each shape primitive independently, the visualizer groups primitives by their parent link as defined in the hierarchical structure and assigns a unique, perceptually uniform color (generated in HSV space) to each link. This simplifies the visual complexity and allows the Geometry Checker to refer to specific links conveniently. Using a pyrender EGL backend for headless operation, it generates four canonical multi-view images with quaternion-based camera positioning to ensure comprehensive object coverage.

Concurrently, a colored point cloud is sampled from the meshes for 3D VLM analysis. This involves a robust process of proportional sampling, allocating points based on the relative surface area of each link to ensure smaller links are not underrepresented, followed by farthest point sampling to guarantee uniform spatial coverage. This converter maintains color consistency with the rendered images by using the identical link-to-color mapping, enabling cross-modal alignment. All generated outputs undergo a unified coordinate normalization process—centering the object at the origin and scaling it to a unit sphere while preserving aspect ratios—to ensure consistency for the downstream Checker modules. The entire pipeline is optimized for the iterative refinement loop, using techniques such as connected component analysis with caching for mesh splitting, parallel rendering, and vectorized NumPy operations for point cloud generation, achieving an end-to-end latency suitable for real-time feedback.

### A.7.5 GEOMETRY CHECKER

The Geometry Checker is a crucial component of our iterative refinement loop, designed to correct geometric errors and physical implausibilities in the initial code generated by the Geometry Coder. This module is powered by a dual-modality system of 2D and 3D Vision-Language Models (VLMs), specifically Gemini-2.5-flash and PointLLM, which provide complementary visual and structural analysis. The Geometry Visualizer first renders multi-view images and a colored point cloud of the object, assigning a unique color to each link. The 2D VLM then analyzes these rendered images from four canonical viewpoints. It leverages the color-to-link mapping to provide precise, localized feedback, such as identifying misaligned components. To ensure this feedback is actionable, the system uses structured extraction and iteration-adaptive prompting that becomes progressively stricter, guiding the Geometry Coder to make targeted corrections.

To detect geometric issues invisible in 2D projections, such as internal intersections or minor disconnections, the 3D VLM analyzes the colored point cloud. This process uses link-proportional sampling, which allocates points based on component surface area to ensure that small but critical parts like hinges are adequately represented. The feedback is structured into a JSON schema with severity-tagged issues (e.g., CRITICAL, MAJOR, MINOR) and requires geometric evidence for each detected fault, significantly reducing false positives. The combined feedback from both 2D and 3D checkers is prioritized based on confidence and severity scores, with critical structural flaws forcing a regeneration cycle. This multi-modal validation ensures that the system corrects for common failures—including floating components, penetrating geometries, and scale inconsistencies—resulting in a final link set that is both visually coherent and physically plausible.

### A.7.6 ARTICULATION CODER

As a core component of the Articulation Builder, the Articulation Coder is responsible for defining the kinematic joint set $\mathcal{J}$ that enables object motion. Taking the validated shape code from the geometry generation stage as input, which specifies the set of links $\mathcal{L} = \{L_i = (\mathcal{M}_i, T_i)\}_{i=1}^{N}$, the coder's primary task is to generate executable code defining the complete kinematic structure. It reasons

about the object's components to establish parent-child hierarchies and form a valid kinematic chain, bridging the geometric representation with a functionally correct articulation structure.

The coder determines the essential parameters for each joint $J_{pc}$, including the joint type ($t_{pc}$), position ($p_{pc}$), motion axis ($a_{pc}$), and motion limits ($l_{pc}$). This is achieved by analyzing the spatial relationships and functional affordances of the links based on their geometry and poses. For instance, it infers joint types (e.g., revolute, prismatic) from semantic cues in the initial prompt and geometric analysis of the links' bounding boxes. The coder also calculates the joint's pose ($T_{pc}$) relative to the parent link and defines its motion axis, considering both local geometry and global object semantics to ensure physically plausible movement. This process operates within a closed-loop, multi-agent pipeline. Concurrently, an Articulation Debugger collaborates with the Coder to resolve any syntax or code-level errors, ensuring the generated script is valid. The validated code is then passed to the Articulation Visualizer for simulation and subsequently assessed by the Articulation Checker. The feedback from the Checker guides the Articulation Coder to iteratively refine the code, correcting functional implausibilities until the final joint set $\mathcal{J}$ is confirmed to be physically correct and well-defined.

## A.8 COST & TIME ANALYSIS

Table 4: Price Comparisons

| Model | Input Price ($) | Cached Input ($) | Output Price ($) |
|---|---|---|---|
| *OpenAI* | | | |
| gpt-5 | $1.25 | $0.125 | $10.00 |
| gpt-5-mini | $0.25 | $0.025 | $2.00 |
| gpt-4o | $2.50 | $1.25 | $10.00 |
| o3 | $2.00 | $0.50 | $8.00 |
| o3-pro | $20.00 | — | $80.00 |
| o1 | $15.00 | $7.50 | $60.00 |
| o1-pro | $150.00 | — | $600.00 |
| *Google* | | | |
| gemini-2.5-pro | $1.25 | — | $10.00 |
| gemini-2.5-flash | $0.30 | — | $2.50 |
| *Anthropic* | | | |
| Claude Opus 4.1[*] | $15.00 | $1.50 | $75.00 |
| Claude Sonnet 4[*] | $3.00 | $0.30 | $15.00 |
| Claude Haiku 3[*] | $0.25 | $0.03 | $1.25 |

To assess the practical viability and efficiency of the LAM framework, we conducted a detailed cost and time analysis based on a representative run generating 15 complex articulated objects. Our implementation strategically utilizes a combination of models: GPT-4o for the high-level reasoning required by the Link Designer, the cost-effective Gemini 2.5 Flash for the iterative VLM feedback in the Geometry and Articulation Checkers, and the powerful Gemini 2.5 Pro for the precise code generation tasks of the Coders. The total cost for generating 15 objects was $0.99, yielding an average cost of just $0.066 per object. The primary cost driver was the 3D Shape Generation stage, which accounted for 39.1% of the total expense, largely due to the 3-5 VLM feedback iterations required per object. The Articulation Logic stage followed closely, consuming 38.1% of the cost with 2-3 feedback iterations, while the initial Link Structure Generation was the least expensive component at 22.8%.

The total pipeline duration for the 15-object batch was approximately 25 minutes, demonstrating the framework's efficiency. On average, generating a single object took 151.4 seconds, with the majority of the time spent in the Shape Generation (85.4s) and Articulation (45.2s) stages. The initial Linker stage was significantly faster, averaging 20.8 seconds. This performance suggests that while the iterative feedback loops are crucial for quality, they are also the main bottleneck. Projecting these figures, generating a larger batch of 1,000 objects would cost an estimated $66.00.

Furthermore, we can project costs for alternative model configurations to balance performance and expense. For example, if we were to use GPT-4o for generating the linker description and a hypothetical, more powerful model like the conceptualized GPT-5 for generating the codes of shape and articulation, the cost profile would change. Based on initial estimates, such a configuration would result in a total cost of approximately $19.50 for generating 159 articulated objects. This highlights the modularity of the LAM framework, where different AI modules can be swapped to meet varying budget and quality requirements. A comprehensive list of current popular LLMs pricing is available.

### A.9 THE SUMMARY OF LLM MODELS USED FOR EXPERIMENTS

**OpenAI**: gpt-5, gpt-4o, o3, o3-pro https://platform.openai.com/docs/pricing?ft-pricing=standard

**Google**: gemini-2.5-flash, gemini-2.5-pro. Reference official page https://ai.google.dev/gemini-api/docs/pricing

**Anthropic**: claude-opus-4.1, claude-sonnet-4 https://docs.anthropic.com/en/docs/about-claude/models/overview#model-comparison-table

### A.10 DEFINITIONS OF THE SHAPE PRIMITIVES

The LAM framework constructs articulated objects using a comprehensive set of geometric tools from the Three.js library. The process begins with fundamental 3D primitives that serve as building blocks, including `BoxGeometry` for rectangular components, `SphereGeometry`, `CylinderGeometry`, `ConeGeometry` for various curved shapes, `TorusGeometry` for ring-like structures, and `PlaneGeometry` for flat surfaces. For more complex forms, the system supports advanced methods such as creating 3D geometry by extruding 2D shapes along a path (`ExtrudeGeometry`, `TubeGeometry`), generating rotationally symmetric objects (`LatheGeometry`), or defining custom 2D profiles (`ShapeGeometry`, `RingGeometry`).

These generated shapes are then combined and modified using several composition techniques. Primitives are organized into complex, articulated hierarchies using `THREE.Group()` for logical assembly. Geometries can be combined through Constructive Solid Geometry (CSG) boolean operations (union, intersection, subtraction) or merged directly for optimization. Custom 2D profiles for these operations are defined using path-based drawing with `THREE.Shape()` and `THREE.Path()`, which utilize commands like `moveTo()` and `bezierCurveTo()`. Finally, each component is precisely positioned, oriented, and scaled in 3D space using transformations for position, rotation, and scale, as well as direct matrix operations.

**Implementation Example (an example by using text prompt "A Rectangular Wooden Cabinet")**

```javascript
import * as THREE from 'three';

export function createScene() {
    const root = new THREE.Group();
    root.name = 'CabinetArticulatedObject';

    // --- Configuration and Dimensions ---
    const SCALE_FACTOR = 0.1; // 10cm = 1 Three.js unit. This ensures all
        dimensions are >= 0.1 units.

    // Helper function to scale dimensions from cm to Three.js units
    const s = (val) => val * SCALE_FACTOR;

    // Cabinet overall dimensions (based on JSON description of "
        cabinet_frame")
    const cabinetWidth = s(100); // 10 units
    const cabinetHeight = s(90); // 9 units
    const cabinetDepth = s(45);  // 4.5 units

    // Frame element thicknesses for planks
    const frameThickness = s(2); // 0.2 units (e.g., outer planks for
        sides, top, bottom, internal dividers)
    const backPanelThickness = s(1); // 0.1 units (thin back panel)

    // Functional gap between components, e.g., doors/drawer and frame.
        (2mm)
    const targetMinimalGap = s(0.2); // 0.02 units

    // Drawer dimensions (based on JSON description of "top_drawer")
    const drawerHeightLink = s(15); // 1.5 units (from JSON description)
    // FIXING: Adjust drawerFaceWidth to allow for 2mm gaps on each side
        (left & right) within the cabinet's internal opening.
    // Cabinet internal width: cabinetWidth - 2*frameThickness = 10 -
        2*0.2 = 9.6
    // Drawer width: 9.6 (inner width) - 2*targetMinimalGap (for left/
        right gaps) = 9.6 - 2*0.02 = 9.56
    const drawerFaceWidth = s(95.6);  // 9.56 units (for 2mm left/right
        gaps)
    const drawerDepth = s(40); // 4 units (assumed for internal drawer
        box depth)

    // Door dimensions (based on JSON descriptions of "left_door", "
        right_door")
    // FIXING: Adjust doorWidth to allow for 2mm gaps on each side (left
        frame, right frame) and 2mm in the center.
    // Total door opening width: cabinetWidth - 2*frameThickness = 9.6
    // Total gaps needed: targetMinimalGap (left frame) +
        targetMinimalGap (right frame) + targetMinimalGap (center) = 3*
        targetMinimalGap = 3*0.02 = 0.06
    // Total width for two doors = 9.6 - 0.06 = 9.54
    // Each door width = 9.54 / 2 = 4.77
    const doorWidth = s(47.7); // 4.77 units.
    const doorHeight = s(75); // 7.5 units
    const doorThickness = s(2); // 0.2 units

    // Handle dimensions
    // FIXING: Reduce handleCylinderRadius for better proportion from s
        (1) to s(0.5).
    const handleCylinderRadius = s(0.5);   // 0.5cm = 0.05 unit
    const drawerHandleLength = s(20);     // 2 units (no change)
    // FIXING: Increase doorHandleLength for better grab proportion from
        s(10) to s(15).
    const doorHandleLength = s(15);       // 1.5 units
```

```
49     const handleProtrusion = s(2);        // 0.2 units (how far handle
           sticks out from surface)
50     // FIXING: Handle offset from inner edge for doors (average of 2-3cm)
51     const doorHandleInnerOffset = s(2.5); // 2.5cm offset
52
53
54     // --- Cabinet Frame (Main Group: all static, non-articulated parts
           of the cabinet structure) ---
55     // This group's origin is set at the center of its base, so that its
           Y=0 is on the floor.
56     const cabinetFrameGroup = new THREE.Group();
57     cabinetFrameGroup.name = 'cabinet_frame'; // From JSON, this is the
           root and contains static parts
58     root.add(cabinetFrameGroup);
59
60     const cabinetFrameRootYOffset = cabinetHeight / 2; // Offset to place
            the cabinet's base at Y=0
61     cabinetFrameGroup.position.y = cabinetFrameRootYOffset;
62
63     // 1. Bottom Panel (was "frame_bottom_plank")
64     // FIXING: Renamed to 'bottom_panel' as per VLM feedback, replacing '
           frame_bottom_plank'.
65     const bottomPlankGeometry = new THREE.BoxGeometry(cabinetWidth,
           frameThickness, cabinetDepth);
66     const bottomPanelMesh = new THREE.Mesh(bottomPlankGeometry);
67     bottomPanelMesh.name = 'bottom_panel';
68     bottomPanelMesh.position.y = -cabinetHeight / 2 + frameThickness / 2;
69     cabinetFrameGroup.add(bottomPanelMesh);
70
71     // 2. Left Side Plank (one of the "side_panels" sub_assembly)
72     const leftSidePlankGeometry = new THREE.BoxGeometry(frameThickness,
           cabinetHeight, cabinetDepth);
73     const leftSidePlankMesh = new THREE.Mesh(leftSidePlankGeometry);
74     leftSidePlankMesh.name = 'frame_left_side_panel';
75     leftSidePlankMesh.position.x = -cabinetWidth / 2 + frameThickness /
           2;
76     cabinetFrameGroup.add(leftSidePlankMesh);
77
78     // 3. Right Side Plank (the other "side_panels" sub_assembly)
79     const rightSidePlankGeometry = new THREE.BoxGeometry(frameThickness,
           cabinetHeight, cabinetDepth);
80     const rightSidePlankMesh = new THREE.Mesh(rightSidePlankGeometry);
81     rightSidePlankMesh.name = 'frame_right_side_panel';
82     rightSidePlankMesh.position.x = cabinetWidth / 2 - frameThickness /
           2;
83     cabinetFrameGroup.add(rightSidePlankMesh);
84
85     // 4. Top Surface (explicitly named "top_surface" in JSON)
86     const topSurfaceGeometry = new THREE.BoxGeometry(cabinetWidth,
           frameThickness, cabinetDepth);
87     const topSurfaceMesh = new THREE.Mesh(topSurfaceGeometry);
88     topSurfaceMesh.name = 'top_surface';
89     topSurfaceMesh.position.y = cabinetHeight / 2 - frameThickness / 2;
90     cabinetFrameGroup.add(topSurfaceMesh);
91
92     // 5. Back Panel (explicitly named "back_panel" in JSON)
93     const backPanelWidth = cabinetWidth - 2 * frameThickness;
94     const backPanelHeight = cabinetHeight - 2 * frameThickness;
95     const backPanelGeometry = new THREE.BoxGeometry(backPanelWidth,
           backPanelHeight, backPanelThickness);
96     const backPanelMesh = new THREE.Mesh(backPanelGeometry);
97     backPanelMesh.name = 'back_panel';
98     backPanelMesh.position.z = -cabinetDepth / 2 + backPanelThickness /
           2;
99     cabinetFrameGroup.add(backPanelMesh);
```

```
100
101    // 6. Horizontal Divider (internal frame structure below the drawer)
102    const horizontalDividerWidth = cabinetWidth - 2 * frameThickness; //
           Spans between side panels
103    const horizontalDividerDepth = cabinetDepth - backPanelThickness; //
           Accounts for back panel
104    const horizontalDividerGeometry = new THREE.BoxGeometry(
           horizontalDividerWidth, frameThickness, horizontalDividerDepth);
105    const horizontalDividerMesh = new THREE.Mesh(
           horizontalDividerGeometry);
106    horizontalDividerMesh.name = 'frame_horizontal_divider';
107
108    // FIXING: Y-position adjustment for horizontal divider to properly
           define the drawer compartment.
109    // The top of the drawer compartment is defined by the bottom of the
           top plank, minus a minimal gap.
110    const drawerCompartmentTopY = topSurfaceMesh.position.y -
           frameThickness / 2 - targetMinimalGap;
111    // The top surface of this divider should be 'drawerHeightLink' below
            drawerCompartmentTopY, minus another gap, and accounting for its
            own thickness.
112    horizontalDividerMesh.position.y = drawerCompartmentTopY -
           drawerHeightLink - targetMinimalGap - frameThickness / 2;
113    horizontalDividerMesh.position.z = 0; // Centered depth-wise for the
           inner space
114    cabinetFrameGroup.add(horizontalDividerMesh);
115
116
117    // --- Top Drawer (Articulated Group) ---
118    const topDrawerGroup = new THREE.Group();
119    topDrawerGroup.name = 'top_drawer'; // From JSON
120    root.add(topDrawerGroup);
121
122    // FIXING: Y-position, X-position, Z-position for 'top_drawer'
           adjusted to remove floating gap and be flush.
123    // Drawer slot bounding Y coordinates within cabinetFrameGroup's
           local system:
124    const drawerSlotTopY = drawerCompartmentTopY; // Already calculated
           for minimal gap below top surface
125    const drawerSlotBottomY = horizontalDividerMesh.position.y +
           frameThickness / 2 + targetMinimalGap; // Top of horizontal
           divider + minimal gap
126
127    const drawerCenterY_relativeToCabinetFrameCenter = (drawerSlotTopY +
           drawerSlotBottomY) / 2;
128
129    // Z-position for top_drawer group: Aligns its local Z=0 (where
           drawer face front will be) with the cabinet's front.
130    const drawerGroupZ_frontFlush = cabinetDepth / 2;
131
132    topDrawerGroup.position.set(
133        0, // Centered horizontally
134        cabinetFrameRootYOffset +
               drawerCenterY_relativeToCabinetFrameCenter, // Global Y
               position to center it in its slot
135        drawerGroupZ_frontFlush // Global Z position so its front surface
                is flush
136    );
137
138    // 1. Drawer Face (explicitly named "drawer_face" in JSON)
139    const drawerFaceGeometry = new THREE.BoxGeometry(drawerFaceWidth,
           drawerHeightLink, frameThickness);
140    const drawerFaceMesh = new THREE.Mesh(drawerFaceGeometry);
141    drawerFaceMesh.name = 'drawer_face';
```

```
142    // Positioned relative to its parent group. Its front surface is
           placed at Z=0 of the group (which is `cabinetDepth/2` globally).
143    drawerFaceMesh.position.set(0, 0, -frameThickness / 2);
144    topDrawerGroup.add(drawerFaceMesh);

145
146    // 2. Drawer Body (sides, back, bottom to make it a physically
           plausible drawer box)
147    const drawerInnerWallThickness = s(1); // 0.1 units for inner drawer
           box planks
148    const drawerInternalWidth = drawerFaceWidth - 2 *
           drawerInnerWallThickness; // Adjusted for new drawerFaceWidth
149    const drawerInternalHeight = drawerHeightLink -
           drawerInnerWallThickness; // Accommodate bottom
150    const actualDrawerBoxDepth = drawerDepth;
151    // Center Z of the internal box relative to `topDrawerGroup` Z=0 (
           cabinet front).
152    const innerDrawerBodyZ = -frameThickness / 2 - actualDrawerBoxDepth /
            2;

153
154    // Drawer Sides (left and right interior panels)
155    const drawerSideGeometry = new THREE.BoxGeometry(
           drawerInnerWallThickness, drawerInternalHeight,
           actualDrawerBoxDepth);
156    const drawerLeftPanel = new THREE.Mesh(drawerSideGeometry);
157    drawerLeftPanel.name = 'drawer_left_panel';
158    drawerLeftPanel.position.set(-drawerFaceWidth / 2 +
           drawerInnerWallThickness / 2, 0, innerDrawerBodyZ);
159    topDrawerGroup.add(drawerLeftPanel);

160
161    const drawerRightPanel = new THREE.Mesh(drawerSideGeometry);
162    drawerRightPanel.name = 'drawer_right_panel';
163    drawerRightPanel.position.set(drawerFaceWidth / 2 -
           drawerInnerWallThickness / 2, 0, innerDrawerBodyZ);
164    topDrawerGroup.add(drawerRightPanel);

165
166    // Drawer Back (interior panel)
167    const drawerBackGeometry = new THREE.BoxGeometry(drawerInternalWidth,
            drawerInternalHeight, drawerInnerWallThickness);
168    const drawerBackPanel = new THREE.Mesh(drawerBackGeometry);
169    drawerBackPanel.name = 'drawer_back_panel';
170    drawerBackPanel.position.set(0, 0, innerDrawerBodyZ -
           actualDrawerBoxDepth / 2 + drawerInnerWallThickness / 2);
171    topDrawerGroup.add(drawerBackPanel);
172
173    // Drawer Bottom (interior panel)
174    const drawerBottomGeometry = new THREE.BoxGeometry(
           drawerInternalWidth, drawerInnerWallThickness,
           actualDrawerBoxDepth);
175    const drawerBottomPanel = new THREE.Mesh(drawerBottomGeometry);
176    drawerBottomPanel.name = 'drawer_bottom_panel';
177    drawerBottomPanel.position.set(0, -drawerInternalHeight / 2 +
           drawerInnerWallThickness / 2, innerDrawerBodyZ);
178    topDrawerGroup.add(drawerBottomPanel);
179
180    // 3. Drawer Handle (explicitly named "drawer_handle" in JSON)
181    const drawerHandleGeometry = new THREE.CylinderGeometry(
           handleCylinderRadius, handleCylinderRadius, drawerHandleLength,
           12);
182    const drawerHandleMesh = new THREE.Mesh(drawerHandleGeometry);
183    drawerHandleMesh.name = 'drawer_handle';
184    drawerHandleMesh.rotation.z = Math.PI / 2; // Rotate to be horizontal
185    // FIXING: Z-position adjusted to be precisely flush with `
           drawer_face` front.
186    // The handle's back surface should align with the drawer face's
           front surface.
```

```
187        // Y-position is already 0, which is vertically centered on the
               drawer face, as requested by VLM.
188        drawerHandleMesh.position.set(
189            0, // Centered X on drawer face
190            0, // Centered Y on drawer face
191            handleCylinderRadius // Positioned so its back is flush with
                   drawer face (front).
192        );
193        topDrawerGroup.add(drawerHandleMesh);
194
195
196        // --- Left Door (Articulated Group) ---
197        const leftDoorGroup = new THREE.Group();
198        leftDoorGroup.name = 'left_door'; // From JSON
199        root.add(leftDoorGroup);
200
201        // FIXING: Y-position: Calculate vertical center for door opening,
               including gaps.
202        const doorOpeningBottomY = bottomPanelMesh.position.y +
               frameThickness / 2 + targetMinimalGap; // Top of bottom panel +
               minimal gap
203        const doorOpeningTopY = horizontalDividerMesh.position.y -
               frameThickness / 2 - targetMinimalGap; // Bottom of horizontal
               divider - minimal gap
204        const doorOpeningCenterY_relativeToCabinetFrameCenter = (
               doorOpeningBottomY + doorOpeningTopY) / 2;
205
206        // FIXING: Hinge at the inner left cabinet edge, offset by
               targetMinimalGap for spacing between door and frame.
207        const leftDoorHingeX = -cabinetWidth / 2 + frameThickness +
               targetMinimalGap;
208
209        // Z-position: Aligns the group's Z origin with the front of the
               cabinet.
210        const doorFrontZ = cabinetDepth / 2;
211
212        leftDoorGroup.position.set(
213            leftDoorHingeX, // Pivot at the inner left edge of the cabinet
                   frame, accounting for slot gap.
214            cabinetFrameRootYOffset +
                   doorOpeningCenterY_relativeToCabinetFrameCenter, // Global Y
                   position to center it in its compartment.
215            doorFrontZ // Global Z position so its front surface is flush.
216        );
217
218        // 1. Left Door Panel (the main part of "left_door" from JSON)
219        const leftDoorPanelGeometry = new THREE.BoxGeometry(doorWidth,
               doorHeight, doorThickness);
220        const leftDoorPanelMesh = new THREE.Mesh(leftDoorPanelGeometry);
221        leftDoorPanelMesh.name = 'left_door_panel';
222        // Positioned relative to its parent group ('leftDoorGroup').
223        // Since the group's origin is the left hinge, the panel extends to
               the right.
224        // The panel's left edge is at the group's origin (hinge). Its center
                is at doorWidth/2.
225        leftDoorPanelMesh.position.set(
226            doorWidth / 2, // Center of panel is at half its width from the
                   hinge (group origin)
227            0, // Centered vertically within group
228            -doorThickness / 2 // Back half of the thickness, to make its
                   front face at Z=0 of the group
229        );
230        leftDoorGroup.add(leftDoorPanelMesh);
231
232        // 2. Left Door Handle (explicitly named "left_door_handle" in JSON)
```

```
233        const leftDoorHandleGeometry = new THREE.CylinderGeometry(
              handleCylinderRadius, handleCylinderRadius, doorHandleLength, 12)
                  ;
234        const leftDoorHandleMesh = new THREE.Mesh(leftDoorHandleGeometry);
235        leftDoorHandleMesh.name = 'left_door_handle';
236        // FIXING: Y-position adjusted to a "more ergonomic" height: 45cm
              from the bottom edge of the door.
237        const newHandleY = -doorHeight / 2 + s(45);
238        // FIXING: X-position consistently "near the edge", 2.5cm from the
              inner (right) vertical edge of the left door.
239        // Door panel extends from 0 to doorWidth in local X. Inner edge is
              at doorWidth.
240        leftDoorHandleMesh.position.set(
241           doorWidth - doorHandleInnerOffset,
242           newHandleY,
243           handleCylinderRadius // Positioned so its back is flush with door
                  face (front).
244        );
245        leftDoorGroup.add(leftDoorHandleMesh);
246
247
248        // --- Right Door (Articulated Group) ---
249        const rightDoorGroup = new THREE.Group();
250        rightDoorGroup.name = 'right_door'; // From JSON
251        root.add(rightDoorGroup);
252
253        // FIXING: Hinge at the inner right cabinet edge, offset by
              targetMinimalGap for spacing between door and frame.
254        const rightDoorHingeX = cabinetWidth / 2 - frameThickness -
              targetMinimalGap;
255
256        // Y-position: Same as left door.
257        // Z-position: Same as left door.
258        rightDoorGroup.position.set(
259           rightDoorHingeX, // Pivot at the inner right edge of the cabinet
                  frame, accounting for slot gap.
260           cabinetFrameRootYOffset +
                  doorOpeningCenterY_relativeToCabinetFrameCenter, // Global Y
                  position to center it in its compartment.
261           doorFrontZ // Global Z position so its front surface is flush.
262        );
263
264        // 1. Right Door Panel (the main part of "right_door" from JSON)
265        const rightDoorPanelGeometry = new THREE.BoxGeometry(doorWidth,
              doorHeight, doorThickness);
266        const rightDoorPanelMesh = new THREE.Mesh(rightDoorPanelGeometry);
267        rightDoorPanelMesh.name = 'right_door_panel';
268        // Positioned relative to its parent group ('rightDoorGroup').
269        // Since the group's origin is the right hinge, the panel extends to
              the left.
270        // The panel's right edge is at the group's origin (hinge). Its
              center is at -doorWidth/2.
271        rightDoorPanelMesh.position.set(
272           -doorWidth / 2, // Center of panel is at half its width to the
                  left of the hinge (group origin)
273           0, // Centered vertically within group
274           -doorThickness / 2 // Back half of the thickness, to make its
                  front face at Z=0 of the group
275        );
276        rightDoorGroup.add(rightDoorPanelMesh);
277
278        // 2. Right Door Handle (explicitly named "right_door_handle" in JSON
              )
```

```
279    const rightDoorHandleGeometry = new THREE.CylinderGeometry(
           handleCylinderRadius, handleCylinderRadius, doorHandleLength, 12)
           ;
280    const rightDoorHandleMesh = new THREE.Mesh(rightDoorHandleGeometry);
281    rightDoorHandleMesh.name = 'right_door_handle';
282    // FIXING: Y-position adjusted to a "more ergonomic" height (same as
           left door handle).
283    // FIXING: X-position consistently "near the edge", 2.5cm from the
           inner (left) vertical edge of the right door.
284    // Door panel extends from -doorWidth to 0 in local X. Inner edge is
           at -doorWidth.
285    rightDoorHandleMesh.position.set(
286        -doorWidth + doorHandleInnerOffset,
287        newHandleY,
288        handleCylinderRadius // Positioned so its back is flush with door
               face (front).
289    );
290    rightDoorGroup.add(rightDoorHandleMesh);
291
292    return root;
293 }
```

## A.11 MORE EXPERIMENTAL DETAILS AND RESULTS

### A.11.1 MORE DETAILS OF THE USED METRICS

Coverage (COV) **Definition**: Coverage (COV) assesses the diversity of generated samples, indicating how comprehensively the model can represent the range of real-world objects. Higher coverage suggests that the generated samples adequately capture the diversity within the reference dataset.

**Calculation**: For each generated object, its closest object in the real dataset is identified using a predefined distance measure. Coverage is then the fraction of unique real objects matched by at least one generated sample:

The formula is:

$$COV(S_g, S_r) = \frac{|\{\text{argmin}_{Y \in S_r} D(X, Y) | X \in S_g\}|}{|S_r|}$$

where $D(X, Y)$ is the distance between object $X$ and object $Y$ Yang et al. (2019).

In the articulated object context, a high coverage means the model successfully generates diverse structures and movements, minimizing issues like mode collapse. The typical distance measure used here is Instantiation Distance (ID).

Minimum Matching Distance (MMD) **Definition**: Minimum Matching Distance (MMD) measures the quality or realism of the generated samples by comparing them to the ground truth set Yang et al. (2019). It calculates, on average, how close each ground truth object is to its nearest neighbor in the generated set Liu et al. (2024b); Yang et al. (2019). A lower MMD indicates that the generated objects are, on average, more similar to real objects, implying higher fidelity Liu et al. (2024b).

**Calculation**: For each reference object $Y \in S_r$, the distance $D(X, Y)$ to its closest generated object $X \in S_g$ is found. The MMD is the average of these minimum distances over all objects in the reference set $S_r$ Yang et al. (2019).

The formula is:

$$MMD(S_g, S_r) = \frac{1}{|S_r|} \sum_{Y \in S_r} \text{min}_{X \in S_g} D(X, Y)$$

where $D(X, Y)$ is the distance between object $X$ and object $Y$ Yang et al. (2019).

When evaluating articulated objects, MMD assesses the realism of the generated part geometries and their articulation parameters Liu et al. (2024b). A low MMD score, using ID or AID as the distance $D$, suggests that the model generates articulated objects whose shapes and motions closely resemble those in the ground truth set Liu et al. (2024b).

1-Nearest Neighbor Accuracy (1-NNA) **Definition**: 1-Nearest Neighbor Accuracy (1-NNA) is a metric used to assess the similarity between the distributions of the generated set $S_g$ and the reference set $S_r$ Yang et al. (2019). It employs a 1-NN classifier to determine if it can distinguish samples from $S_g$ versus $S_r$ based on their nearest neighbors in the combined set Yang et al. (2019). If the two distributions are identical, the 1-NNA should be close to 50% (chance level) Yang et al. (2019). Deviations from 50% indicate discernible differences between the distributions. Thus, a score closer to 50% is better, suggesting that the generated distribution is a good approximation of the true data distribution Yang et al. (2019).

**Calculation**:

1. Combine the generated set $S_g$ and the reference set $S_r$ into a single dataset $S_{all} = S_g \cup S_r$.

2. For each sample $Z \in S_{all}$, find its nearest neighbor $N_Z$ in $S_{all} - \{Z\}$ using a distance metric $D$.

3. The sample $Z$ is classified as "generated" if $N_Z \in S_g$ and "real" if $N_Z \in S_r$.

4. 1-NNA is the accuracy of this classification: the proportion of samples whose predicted label (based on their nearest neighbor's origin) matches their true origin Yang et al. (2019).

The formula is:

$$1 - NNA(S_g, S_r) = \frac{\sum_{X \in S_g} \mathbb{I}[N_X \in S_g] + \sum_{Y \in S_r} \mathbb{I}[N_Y \in S_r]}{|S_g| + |S_r|}$$

where $\mathbb{I}[\cdot]$ is the indicator function, and $N_X$ (or $N_Y$) is the nearest neighbor of $X$ (or $Y$) in $(S_g \cup S_r) - \{X \text{ or } Y\}$ Yang et al. (2019). An ideal score is 0.5 (or 50%).

For articulated objects, 1-NNA provides a measure of how well the overall distribution of generated shapes and articulations matches the ground truth distribution Liu et al. (2024b). It considers both the quality (similarity to individual real objects) and diversity (coverage of the true distribution's modes) Yang et al. (2019). The CAGE paper reports 1-NNA using Abstract Instantiation Distance (AID) as the distance metric Liu et al. (2024b). A 1-NNA score closer to 50% indicates that the generated articulated objects are hard to distinguish from real ones distributional.

Ensuring that the generated 3D scene aligns with the input text prompt is crucial for text-based scene generation methods. We assess this controllability using the following established metrics:

CLIP Score **Definition**: The CLIP (Contrastive Language-Image Pre-training) Score measures the semantic alignment between an image and its corresponding text description. It calculates the cosine similarity between the image embedding and text embedding derived from the CLIP model. Higher scores reflect better semantic consistency between the visual content and textual prompt.

**Usage**: Within the domain of 3D scene generation, the CLIP Score quantitatively assesses how closely the rendered images from a generated 3D scene match the semantic content specified in the input textual description. It serves as an objective metric for evaluating the fidelity of generated scenes in capturing the intended textual semantics.

BLIP Score **Definition**: The BLIP (Bootstrapping Language-Image Pre-training) Score evaluates the correspondence between an image and its caption. Specifically, it employs the Image-Text Matching (ITM) head from the BLIPv2 model, which classifies image-text pairs as either matching or non-matching. A higher BLIP score indicates a stronger image-text relationship.

**Usage**: Analogous to the CLIP Score, the BLIP Score is utilized to measure how well the generated 3D scene aligns visually with the provided textual prompt. It provides complementary insights into the controllability and semantic accuracy of the generated outputs.

## A.12 MORE VISUALIZATIONS

### A.12.1 ANALYSIS OF FAILURE CASES

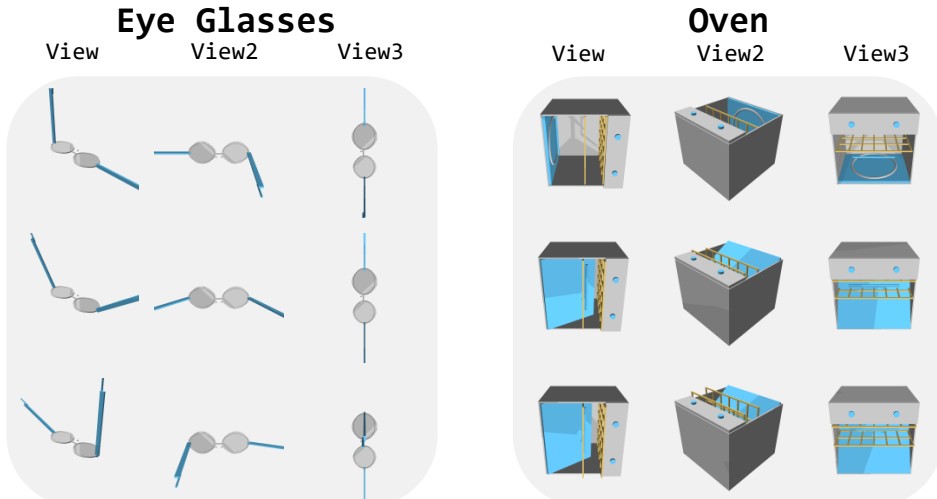

Figure 10: Qualitative comparison of articulated object generation from text prompts.

Figure 10 from the supplementary material highlights the comparative strengths of the iArt model in generating articulated objects, demonstrating notable improvements over existing methods like Singapo and Articulate Anything. Examples such as the "Pliers," "Door," and "USB" illustrate that iArt can produce coherent structures with plausible articulations. Nonetheless, generating accurately articulated 3D objects remains inherently challenging. Beyond correctly forming individual parts, the model must precisely capture complex kinematic relationships and constraints between these parts. Even when part geometry is acceptable, subtle inaccuracies often occur in defining joints, particularly for objects featuring multiple degrees of freedom or uncommon articulation mechanisms.

Ensuring perfect articulation, especially the precise orientation of joint axes and the accurate range and direction of movement, continues to pose significant difficulties. For example, complex objects like the multi-joint "Lamp" or the "Faucet" generated by iArt might appear structurally sound in static images. However, precisely controlling each rotation axis and maintaining realistic motion limits is intricate. An incomplete or partially incorrect interpretation of the object's functional design might cause joints to be assigned plausible yet practically inaccurate rotational directions or axes. Despite significant advancements shown by models such as iArt, accurately interpreting and implementing nuanced joint orientations and movements remains a challenging area requiring further refinement.

### A.12.2 MORE COMPARISONS WITH PREVIOUS WORKS

Figure 11 illustrates a qualitative comparison of our method against Singapo and Articulate Anything across nine object categories (Cart, Chair, Door, Faucet, Lamp, Lighter, Pliers, Camera, and USB). Our approach, iArt, consistently generates more recognizable, coherent, and accurately articulated 3D objects. For instance, where Singapo often produces jumbled or abstract forms and Articulate Anything may result in disconnected or simplistic representations, our method successfully yields well-defined structures like complete carts, realistic chairs, and identifiable faucets with distinct components. This visual evidence underscores our method's superior capability in capturing essential geometry and articulation from text, leading to more realistic and functionally plausible models across a diverse set of objects.

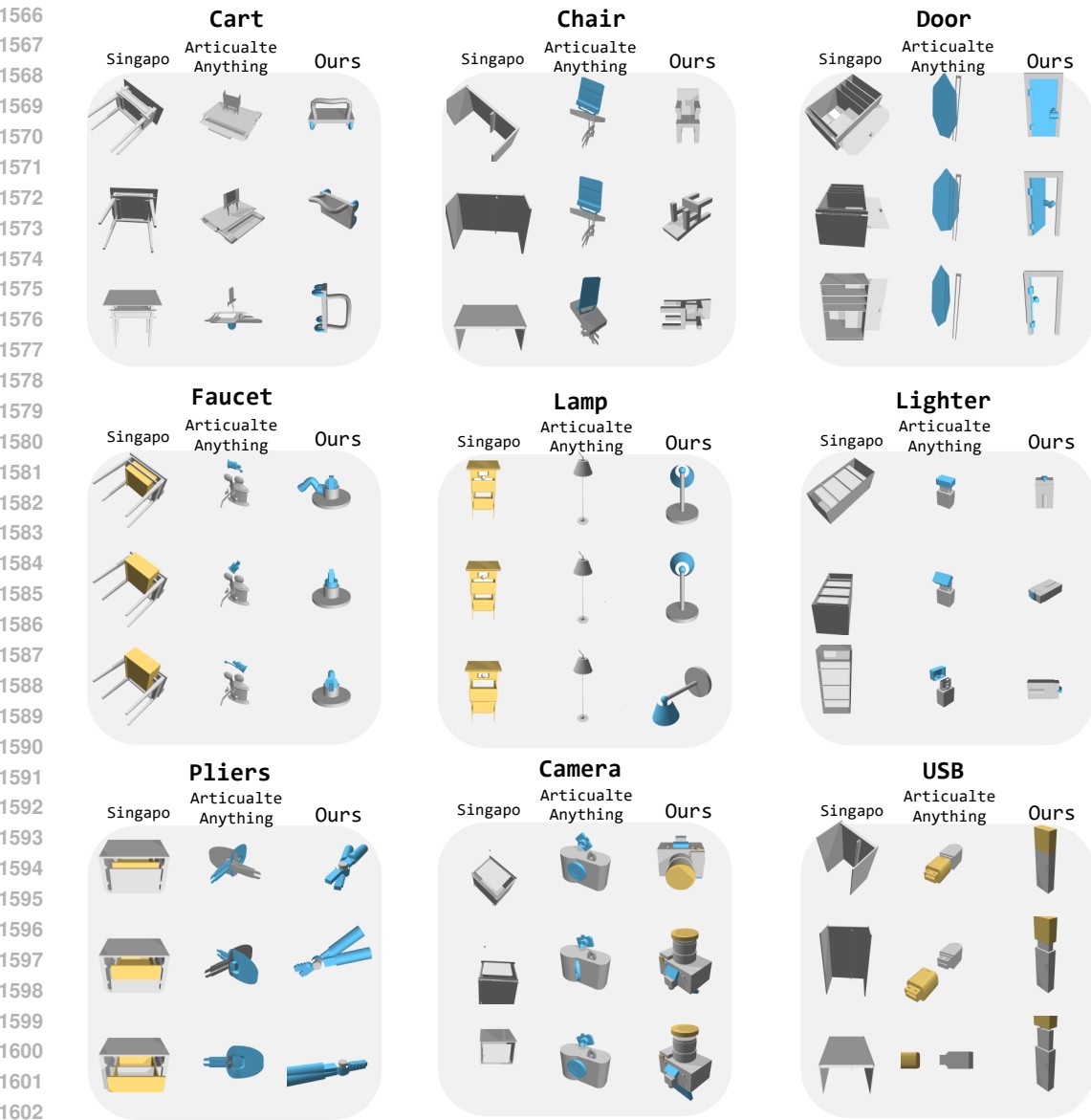

Figure 11: Qualitative comparison of articulated object generation from text prompts.

