# OpenReview forum: "LAM: Language Articulated Object Modelers"
_ICLR.cc/2026/Conference — ICLR 2026 Conference Withdrawn Submission_

### Official Review · Reviewer_uJeV · 2025-10-22

**Soundness:** 2
**Presentation:** 2
**Contribution:** 2
**Rating:** 2
**Confidence:** 3

**Summary:**

The paper introduces LAM, a novel system for generating articulated 3D objects from text prompts. Its core innovation is framing the entire process as a unified code generation task, where both geometry and articulation are co-designed from scratch. LAM employs a multi-agent framework of LLMs and VLMs. A Link Designer first creates a hierarchical part structure from the text. Coders then write executable code to define part geometries and kinematic joints. Crucially, the system uses a closed-loop, multi-modal feedback mechanism: VLM-powered Checkers analyze rendered images and point clouds of the generated object, providing critiques that guide the Coders to iteratively

**Strengths:**

1: The central idea of reframing articulated object generation as a code generation task is exceptionally strong. This avoids the ambiguity and imprecision of directly generating meshes or voxels.

2: The iterative self-correction loop is the system's most impressive technical contribution.  It successfully mimics a human design-and-critique workflow.

**Weaknesses:**

1: The system is a complex integration of multiple (and usually proprietary) models (GPT-4o, Gemini, PointLLM). This makes the research difficult to reproduce, expensive to run, and blurs the lines of which specific component is responsible for which capability. It feels more like a sophisticated systems paper or an engineering one rather than a single, self-contained novel model.

2: The system's creative ceiling is defined by its predefined library of geometric primitives and CSG operations.  This is similar to the use of blender or other procedural rule-based modeling tools.  I cant see clear advantages.

**Questions:**

How might this framework be extended to handle free-form shapes beyond the current library of geometric primitives?

Will the whole system/pipeline benefits from fine-tuned llms and vlms? Like using llama/qwen then fine-tune with some samples.

---

### Official Review · Reviewer_2qst · 2025-10-23

**Soundness:** 2
**Presentation:** 1
**Contribution:** 3
**Rating:** 2
**Confidence:** 4

**Summary:**

The paper introduces a method called LAM to create objects using only text that have parts to them which can be moved. These movable parts need to have correct joint and link placements between each other. LAM is an iterative approach, which can refine its initial predictions through textual feedback. The method leverages many different LLMs and 2D/3D VLMs and consists of 3 main parts. First, the Link Designer which creates a tree consisting of the different low-level parts of the object. Second, the Articulable Geometry Coder, which builds/codes the object itself. Third, the Articulation Builder, which builds/codes the movements possible with the object. The second and third step both use a VLM-based Checker at the end, which can be used to refine the generation in an iterative way. The method is evaluated on text-based joint prediction, visual alignment, articulation modeling and in-distribution generation quality. LAM is compared to different baseline methods, which also only use text-based input for generation.

**Strengths:**

**Originality**
- The idea of combining different foundation models in a collaborative system with an internal feedback loop is novel in this respective field, as well as the simplicity to only needing text input to create articuable objects
- LAM is not only evaluated on pre-set objects, but also on open-vocabulary scenarios and instruction-following ability, which provides a better insight into the possibilities of the model

**Quality**
- The high-level method is explained in detail and visualized with easy to understand figures
- The paper as a whole is mostly well written
- The appendix includes a broad range of additional information

**Clarity**
- The structure of the paper is clear
- Sentences are mostly easy to understand and to follow along
- The method itself is straight forward

**Significance**
- Creating articuable objects through a single text prompt can help to increase dataset sizes and produce new data, which for example can be used for robot learning
- The method shows an increase in performance compared to other methods which only use text as an input, making LAM more reliable in the correct generation of articulable objects

**Weaknesses:**

**Motivation of the Paper**
- The overall motivation of the paper is not clear. It is understandable that there is a need for more articuable objects, but not why it has to be as text-form input instead of images or videos. The following sentence from line 75-77 should therefore be reworked and the need for textual descriptions should be made more clear.
   - "Automating the generation of articulation-ready models from textual descriptions represents a promising approach that we explore here to address this gap and enhance scalability in the creation of interactive virtual environments."
- With regards to different inputs for articuable object generation models, it is not clear why images or videos limit the diversity and constrain the models. Rewriting line 82-84 would help to clarify what exactly limits these models diversity and how text input improves this. Citations or additional experiments would be helpful to strengthen this claim.
   - "However, these methods are constrained by their reliance on structured data as input, which limits the diversity of producible articulated objects."


**Paper Structure/Formulations**
- Figure 1 is unnecessary, as it does not serve a purpose, but to show generated objects (which can be seen later on in the experiment section). It also takes away a lot of space, which could be used to put important information (like mentioning specific LLM/VLM models or more information on the modules) from the appendix into the main paper. Additionally, mentioning the object generation with texture is only ever again referenced in the appendix and not in the main paper.
- The contribution claim (2) is part of the contribution (1) and contribution (3) are the experiments conducted to verify the method, which in itself should not be considered a contribution, but is necessary to prove the performance of LAM. I would suggest to merge contribution (1) and (2) and leave out (3) to make one contribution for the whole paper.
- Chapter 3.2 and 3.3 would benefit a lot from more specific information, like what LLMs/VLMs were used, what the output types are and a more detailed description of the different modules. This information is partly present in the appendix, but is not frequently referenced and the main paper should stand alone without appendix information.
    - It was, for example, also not clear that the Geometry Coder and Articulation Coder are LLMs as it was not mentioned in this chapter
- There are missing citations/sources for different claims or missing experiments in the paper
   - Line 189-190: Direct URDF generation is inherent oversimplified
   - Line 202-204: How does LAM or text input give more control? This is not clear from the sentences and there are no experiments regarding this claim in the paper.


**Experiments/Ablations and Metrics**
- The paper would greatly benefit from more ablations
    - Chapter 4 does not discuss why LAM uses different LLMs and VLMs for the different steps, neither are there ablations showcasing different results using different LLMs and VLMs
   - Table 3 only considers 1 and 3 iterations. What happens with 2 or more then 3?
- There are missing downstream tasks that showcase the usefulness of having more accurately generated articuable objects
- Metrics which were used are not evaluated on all models present, but specific metrics only use specific models. This seems to be the case, because the results are taken from those specific papers. It would be helpful if the authors could clarify what models were actually re-implemented and evaluated by themselves and which take results from the respective paper.
   - Why were models only evaluated on the joint prediction metric and not the link placement metric, as described in Articulate Anything?
   - Results from CAGE are taken from ArtFormer and not from the CAGE paper itself. Please clarify why?
   - There are also missing evaluations regarding the costs in USD of LAM vs. the baselines, which would be interesting additional information, as using SOTA LLMs and VLMs can be very costly


**Minor Concerns**
- Using only closed-weight LLMs/VLMs restricts the usage of LAM if you have no access to such models and it would be interesting to see how open-weight models perform
- Figure 3 is never mentioned in the text
- LAM is never introduced as a acronym in the text
- Line 52-53: "remains a critical bottleneck" - could you provide a source for this claim?
- Spelling/Grammar:
    - Line 36: "The LAM first reasons..." should be reason
    - Links as a term is introduced more then 3 times (do it once)
    - Related Works subsection titles - Is the s necessary behind object?
- Chapter 3 should not be an acronym and better be named “Method”, there is also text missing as it is an empty chapter
- Line 161, what does LL stand for?
- Citation missing for Three.js library in line 191 and 192
- Naming of the single parts in the method section are very confusing to read in text
    - Remove the Geometry and Articulation part and only call the single parts Coder, Debugger, etc. This makes following the text much easier
- What exactly are the CLIP and BLIP scores? In the paper it is reported that it is their cosine similarity, which should be between -1 and 1. Why are values much bigger? What do they mean in this context?
- Table 2 a): The LAM result for BLIP should be bold
- Line 395 there is an “I” used in the text, should maybe be a "we"?
- Please also add Limitations and Future Work to Chapter 5 or reference it in the appendix
- Add the citation references to the baseline methods in the tables

**Questions:**

**Questions**
- What is the exact motivation of the paper? Why is text-based input superior to any other input for such generation models? How does your model make the output more diverse?
- Why did you not consider open-weight foundation models?
- How did you decide for your evaluation metrics and what baseline models did you implement/evaluate yourself?

**Suggestions**
- Rewrite your paper to make the motivation and method part clearer
- Strengthen your claims with appropriate citations, especially when motivating LAM
- Rework the structure of the paper to add space for more information from the appendix
- Add more ablations to showcase why you choose the specific implementations
    - Any additional ablations you are able to do during the rebuttal period would greatly benefit the paper

---

### Official Review · Reviewer_2jsW · 2025-10-25

**Soundness:** 2
**Presentation:** 2
**Contribution:** 3
**Rating:** 4
**Confidence:** 3

**Summary:**

LAM generates code for articulated object synthesis from natural-language text prompts, without relying on visual inputs or external assets. The framework consists of three key modules: 1) Link Designer parses the text prompt into object link layouts. 2) Geometry Coder converts text-described parts into executable code, and self-evaluates the plausibility of the generated code and rendered shapes. 3) Articulation Builder predicts the joints between the generated shapes and self-assesses their correctness.

**Strengths:**

-   The idea of employing LLMs to generate code that defines articulated objects using predefined shape templates is novel.
-   Experiments across multiple tasks demonstrate the effectiveness of the proposed method.
-   The generated articulated objects exhibit high structural complexity.

**Weaknesses:**

-   Presentation
    -   The naming of modules is confusing. In Fig.3, the overall pipeline includes "Articulable Geometry Coder" as a key module, but Fig.4 instead shows "Articulation Shape Coder," with "Geometry Coder" as a subcomponent.
    -   Several typos exist, *e.g.*, "LL" in Line 161 and "Articualte" in Fig. 11.
    -   In consistent notation. \mathcal{M} is used in Line-143 to define L_{i}, but a plain M is used in Line-154 to describe L_{i}.
    -   In Sec.4.1, the "Joint Prediction" experiment involves comparisons across 5 categories and "general classes." However, another paragraph titled "Comparisons on General Classes" at the end of Sec.4.1 refers to a different experiment, which is confusing.
-   Method
    -   Details are missing on how gpt-4o is prompted to parse link layouts in the Link Designer, as well as similar prompting strategies for the Geometry Coder and Articulation Builder.
    -   The generated articulated objects are composed primarily of simple geometric primitives, lacking finer geometric details or outliers commonly found in real-world objects.
-   Experiment
    -   The reference objects used for computing COV, MMD, and 1-NNA are not specified, making it difficult to assess the fairness and validity of the evaluation.
    -   The ablation studies only analyze the effect of the "checker" modules, omitting discussion of the code generators themselves.
-   There is a visible shape error in Fan.mp4 in the supplementary material, where the bell-like meshes at the base of the fan appear inside-out. This raises concern about the effectiveness of Geometry Coder.

**Questions:**

-   Geometry Coder
    -   How does the Geometry Coder determine which shape (*e.g.*, box, cylinder) best fits each part described in the link layout? For example, how is the seat of a chair encoded as a box or a cylinder?
    -   How does it decide when to apply CSG to form more complex shapes?
    -   If a new base geometry is introduced in THREE.js, how can the Geometry Coder adapt to and utilize it?
-   What are the joint origin and axis error rates in the joint prediction results?

---

### Official Review · Reviewer_vDVP · 2025-10-31

**Soundness:** 2
**Presentation:** 3
**Contribution:** 2
**Rating:** 4
**Confidence:** 4

**Summary:**

he paper proposes LAM, an agentic pipeline that turns a text prompt into an articulated 3D object by generating code for both geometry and joints, iteratively debugged and refined using 2D/3D VLM “Checkers.” The system decomposes the object with a Link Designer, writes Three.js-style geometry code and URDF-level articulation code, simulates/visualizes the result, then uses VLM feedback to fix geometry/joint errors.

**Strengths:**

1- Closed-loop multi-modal checking: practical VLM-driven feedback to correct geometry (“legs misaligned”) and articulation (“wrong hinge direction”) is convincingly described.

2- Empirical gains: Big margins on joint prediction success (e.g., 77.1% vs 40.3% vs 13.5% on 5 Real2Code classes; 63.7% vs 48.9% on general classes). Visual/semantic and distributional metrics also improve (best CLIP/BLIP; best MMD/COV/1-NNA).

**Weaknesses:**

1- Novelty vs. close baselines: Prior works already auto-generate code/URDF or retrofit articulations (e.g., Articulate-Anything, Real2Code). The novelty is the unified code representation + multi-agent VLM loop, but the distinction could be argued more sharply and with deeper analysis of why code-as-IR beats graph/mesh priors beyond controllability anecdotes.

2- Evaluation dependence on VLMs: Some judgments (e.g., articulation plausibility and preferences) use GPT-4o; this risks self-evaluation bias and weakens objectivity unless human protocols are extensive and blinded

**Questions:**

1- How do you prevent evaluation leakage when GPT-4o is used as both a Checker in the loop and as an evaluator?

2- What is the failure taxonomy (geometry vs. joint type vs. axis vs. limits vs. motion direction vs. collision), and their relative frequencies pre/post Checkers?

3- What is the upper bound on link/joint count where your approach remains stable and cost-effective?

---

### Note · Authors · 2025-11-13

**Comment:**

Thank you very much for the thoughtful and constructive feedback. We truly appreciate the time and effort you put into reviewing our work.
After careful consideration, we have decided to withdraw the paper at this stage. We will incorporate the valuable suggestions into our revised version and continue improving the work. Thank you again for your professional and helpful comments.

**Withdrawal Confirmation:**

I have read and agree with the venue's withdrawal policy on behalf of myself and my co-authors.